

# Measuring clients' experiences with antenatal care before or after childbirth: it matters

Marisja Scheerhagen[1,2], Erwin Birnie[3], Arie Franx[2], Henk F. van Stel[4,†] and Gouke J. Bonsel[3]

[1] Department of Obstetrics and Gynecology, Erasmus Medical Centre, Rotterdam, The Netherlands
[2] Department of Obstetrics and Gynecology, University Medical Center Utrecht, Utrecht, The Netherlands
[3] Department of Obstetrics and Gynecology, Academic Collaborative Maternity Care, University Medical Center Utrecht, Utrecht, The Netherlands
[4] Julius Center for Health Sciences and Primary Care, Department of Healthcare Innovation and Evaluation, University Medical Center Utrecht, Utrecht, The Netherlands
[†] Deceased.

Corresponding author
Erwin Birnie, e.birnie@umcutrecht.nl

## ABSTRACT

**Background**. When clients' experiences with maternity care are measured for quality improvement, surveys are administered once, usually six weeks or more after childbirth. Most surveys conveniently cover pregnancy, childbirth and postnatal care all in one. However, the validity of measuring the experiences during pregnancy (antenatal experiences) after childbirth is unknown. We explored the relation between the measurement of antenatal experiences late in pregnancy but prior to childbirth ('test' or gold standard) and its retrospective measurement after childbirth (retrospective test). Additionally, we explored the role of modifying determinants that explained the gap between these two measurements.

**Methods and Findings**. Client's experiences were measured by the ReproQuestionnaire that consists of an antenatal and postnatal version, and covers the eight WHO Responsiveness domains. 462 clients responded to the antenatal and postnatal questionnaire, and additionally filled out the repeated survey on antenatal experiences after childbirth. First, we determined the association between the test and retrospective test using three scoring models: mean score, equal or above the median score and having a negative experience. The association was moderate for having any negative experience (absolute agreement = 68%), for the median (absolute agreement = 69%) and for the mean score (ICC = 0.59). Multiple linear and logistic regression analysis for all three scoring models revealed systematic modifiers. The gap between antenatal and postnatal measurement was (partly) associated with clients' experiences during childbirth and postnatal care and by professional discontinuity during childbirth but unrelated to the perceived health outcome.

**Conclusions**. The antenatal experiences should be measured before and not after childbirth, as the association between the antenatal experiences measured before and after childbirth is moderate.

## INTRODUCTION

Clients' experiences with care are considered to be an important independent indicator of health care performance (*Valentine et al., 2003*; *Valentine, Bonsel & Murray, 2007*). Being relevant for its own sake, clients' experiences could also affect health outcome through several pathways (*Campbell, Roland & Buetow, 2000*; *Sitzia & Wood, 1997*; *Wensing et al., 1998*; *Williams, 1994*). For example, clients who truly understand the explanation of their caregiver are more likely to comply to treatment or lifestyle change.

As clients' experiences are an independent indicator of performance, clients' experiences are systematically measured using surveys, usually held after the care-episode. Such measurements could help to identify areas for improvement (*Haugum et al., 2014*; *Weinick et al., 2014*). Targets of quality improvement are found by identifying health care organizations or areas with below average scores or single negative outliers on questions representing the characteristics of service delivery, e.g., communication and prompt access to services. Next, the organization develops and implements a plan to meet these goals, and verifies if the goals are met (*UK Department of Health, 2010*; *Ellis, 2006*; *Ettorchi-Tardy, Levif & Michel, 2012*; *Kay, 2007*).

Clients' experiences in maternity care are routinely measured in several countries. Data on clients' experiences are usually collected through surveys administered six weeks or more after childbirth. Most surveys cover pregnancy, childbirth and postnatal care in one measurement (*Dzakpasu et al., 2008*; *Hay, 2010*; *Redshaw & Heikkila, 2010*; *Van Wagtendonk, Hoek & Wiegers, 2010*; *Wiegers et al., 1996*). As these surveys cover almost about 9 months of care, with different health care professionals, settings and possibly events, measurement of client's experiences bears the risk of being vulnerable to memory failure and/or changes in perception due to modifying intercurrent events that happened since the antenatal experiences. Assuming the antenatal measurement of such experiences to be the gold standard, the question is whether the response on the postnatal survey shows random and/or systematic error. Stated otherwise, when the clients' experiences are measured before childbirth and repeated after childbirth, does this lead to the same clients' experience scores? Ideally, valid measurement of antenatal experiences postnatally should not be systematically affected by the care process, experiences or outcomes that occur *after* antenatal measurement. Despite the widespread practice of a one-stage postnatal measurement, to our knowledge this question has never been explored. If random error is considerable or systematic shifts are present, the convenient one-stage measurement perhaps should be replaced by a two-stage measurement procedure, that includes the measurement of clients' experiences not only after childbirth but also antenatally.

We explored the presence of memory effects in the measurement of clients' experiences in maternity care using the ReproQuestionnaire (ReproQ). ReproQ is the national survey for client experience measurement in childbirth care in the Netherlands. It was especially designed for a two-stage measurement procedure, consisting of antenatal and postnatal versions. ReproQ was extensively validated ($n > 18,000$) (*Scheerhagen et al., 2015*; *Scheerhagen et al., 2016*) and is currently regarded as one of the national maternity care indicators (*CPZ, 2015*).

## METHODS

### ReproQuestionnaire

The ReproQ consists of two versions, each covering the experiences of two reference periods. The antenatal version covers the experiences during early and late pregnancy; the postnatal version covers the experiences during childbirth and postnatal care. Both versions are identical, in the sense that the same type of experiences is asked for, but items (questions) are contextually adapted. Altogether, a client is invited to judge a typical item for four consecutive periods.

The conceptual basis of the ReproQ was the WHO responsiveness model (*Valentine et al., 2003*; *Valentine, Bonsel & Murray, 2007*). The WHO developed this universally applicable concept that consists of four domains on the interactions of the client with the health professional (dignity, autonomy, confidentiality, and communication), and of four domains on the client orientation of the organizational setting (prompt attention, access to family and community support, quality of basic amenities, and choice and continuity of care) (*Valentine et al., 2003*; *Valentine, Bonsel & Murray, 2007*). The response mode of all the experience items uniformly consists of four categories: "never", "sometimes", "often", and "always", with a numerical range of 1 (worst) to 4 (best).

Additional sections of the ReproQ address the client's socio-demographic characteristics, details about the care process during pregnancy and childbirth, and maternal and infant health outcomes in non-medical terms as perceived by the mother. We also added a relevance question on which two out of eight domains were most important to the client.

Previous psychometric analyses showed that content and construct validity were good, as was the test-retest reliability of the experience during childbirth. Full details of the development and the psychometric properties of the questionnaire are described elsewhere (*Scheerhagen et al., 2015*; *Scheerhagen et al., 2016*).

### Design, ReproQ scoring models, outcomes

The Medical Ethical Review Board, Erasmus Medical Center, Rotterdam, the Netherlands, approved the study protocol (study number MEC-2013-455).

The study was designed as a cohort study with three measurements. First, women received an invitation to fill out the antenatal ReproQ around a gestational age of 34 weeks. This is called 'test'. Second, women received an invitation to fill out the postnatal ReproQ six weeks after the expected date of childbirth. Non-responding women received a reminder two weeks after invitation to the antenatal and postnatal questionnaire. Third, we invited women who responded to the antenatal and postnatal ReproQ again to fill out the antenatal experiences after childbirth. This is called the 'retrospective test'. We sent the retrospective test at least 14 days after women filled out the postnatal ReproQ.

Three different scoring models exist to summarize clients' experiences and to monitor adverse outcomes at the individual or aggregate level. The three models may be applied to an individual item, to an individual domain (called domain score), to two summary scores of the four personal and four setting domains (called personal and setting score), or to a summary score of all domains (called total score).

**Table 1  Scoring models, outcome measures and measures of association.**

| Scoring model | Definition | Outcome measure | Measure of association | | |
|---|---|---|---|---|---|
| | | | For summary and domain scores | For item scores | In regression analysis |
| Negative score | Ticking the category 'never' in at least one of the items of a domain (indicating a very poor experience), and/or filling out 'sometimes' in at least one of the items of the two domains that the client identified as most important | Dichotomous | AA | AA | OR |
| Mean score | The unweighted average score of items within a domain, treating the item response categories numerically; the total, personal and setting summary scores equal the mean of the mean domain scores involved in that summary measure | Mean (SD) | ICC | AA | $\beta$ |
| Median score | Whether the client's mean item, domain or summary score is equal to/above or below the median of the distribution of the respective item, domain or summary scores of all cases | Dichotomous | AA | AA | OR |

Table 1 displays the scoring models and their definitions. The first model creates a dichotomous variable (called 'negative score') at the client level, reflecting the presence of any so-called negative experience. As Table 1 shows, the definition of a 'negative' experience is based in part on the two domains that a client identifies as most important, thereby creating a personalized score. Since the likelihood of a negative experience partially depends on the number of items per domain, absolute percentages of negative scores cannot be compared across domains. The negative score model assumes that, for the individual client or for an organisation, a negative experience cannot be compensated by very good experiences on other items or domains. This is contrary to the mean score where good experiences can compensate poor experiences.

The second scoring model computes a continuous mean score (called 'mean score', range 1.0–4.0) at the client level, for each domain or group of domains separately. The total, personal and setting summary scores are not the mean of all items involved in the domains, but the unweighted mean of the mean domain scores involved in that summary measure. For the calculation of the summary scores, each domain has the same weight, even if the domains rest on a different numbers of items.

Finally, the third model creates a dichotomous variable at the client level reflecting whether her mean item, domain or summary score is equal to/above *or below* the median *of the distribution* of the respective item, domain or summary scores *of all cases* (called 'median score'). The 'median score' model was added because of the skewed distributions of clients' experience scores.

## Data collection

ReproQ data were obtained from two sources: 10 perinatal units (a hospital with its associated community midwife practices) and two maternity care organizations. These organizations deliver postnatal care at home from childbirth onwards over a period of seven to 10 days. Women can register and apply for this service during pregnancy. For perinatal units, clients were invited to participate by their caregiver, who asked for consent.

For maternity care organizations, all women were invited to fill out the client experience questionnaire, after consent was ticked.

Data were collected in two periods. In the first period (October 2013 to January 2015), data was collected with the antenatal ('test') and postnatal ReproQ. There were no restrictions to invite women to fill out the antenatal and postnatal ReproQ; all women could participate provided that informed consent was signed or ticked. The second period, December 2014, administered the data of the retrospective test. Women were excluded from participation of the retrospective test for the following reasons: (1) women did not respond to the antenatal and postnatal questionnaires, (2) women filled out less than 50% of the antenatal and/or postnatal experience score, or (3) they filled out the questionnaires on paper. (This was done for the reasons of data management efficiency; $n = 166$). Women were excluded from analyses if they filled out less than 50% of items of the retrospective test questionnaire, or if women filled out the retrospective test over 1.5 years after childbirth. The latter criterion excluded women who could be pregnant again.

## Measures of agreement

In this study we used two dichotomous scores and one continuous score for the domain and summary scores, with two different agreement statistics. For the negative and median scores, we used the percentage absolute agreement (AA), classified as 'excellent' (90%–100%), 'good' (75%–89%), 'moderate' (60%–74%), or 'poor' (<60%) (*Singh et al., 2011*). For the mean score, we used the Intraclass Correlation Coefficient (ICC) as measure of agreement (two way mixed model, absolute agreement, single measure), and classified the estimated ICCs as: 'excellent' ($\geq.81$), 'good' (.61 –.80), 'moderate' (.41–.60), 'poor' ($\leq.40$) (*Singh et al., 2011*). For the individual items, agreement between the test and retrospective test was quantified as the percentage absolute agreement.

## Data analysis

Figure 1 shows the analytic framework. All analyses were performed on the reported experience of the second half of the pregnancy, because in psychometric analysis the experiences during first and second half of pregnancy are highly associated ($AA_{Neg} = 91.6\%$; $AA_{MD} = 85.9\%$; ICC = 0.83). The late antenatal experiences were chosen as comparator ('test' or gold standard), because the second half of pregnancy covers more antenatal check-ups than the first half, and therefore thought to be more representative for the entire antenatal phase. Moreover, the timespan between the second half the pregnancy and the retrospective test is smaller than the timespan between early pregnancy and the retrospective test, and therefore the risk of memory effects is probably smaller.

We used all retrospective test data collected up to 1.5 years after childbirth (range: 3.5 month to 1.5 years after childbirth). The wide range had limited impact on the experience scores of the retrospective test and the association between the test and retrospective test; both slightly decreased over time.

First we explored the crude agreement between the antenatal experiences measured before (test or gold standard) and after childbirth ('retrospective test'). For that purpose the three outcome measures were computed for a. the total score, b. the personal and setting

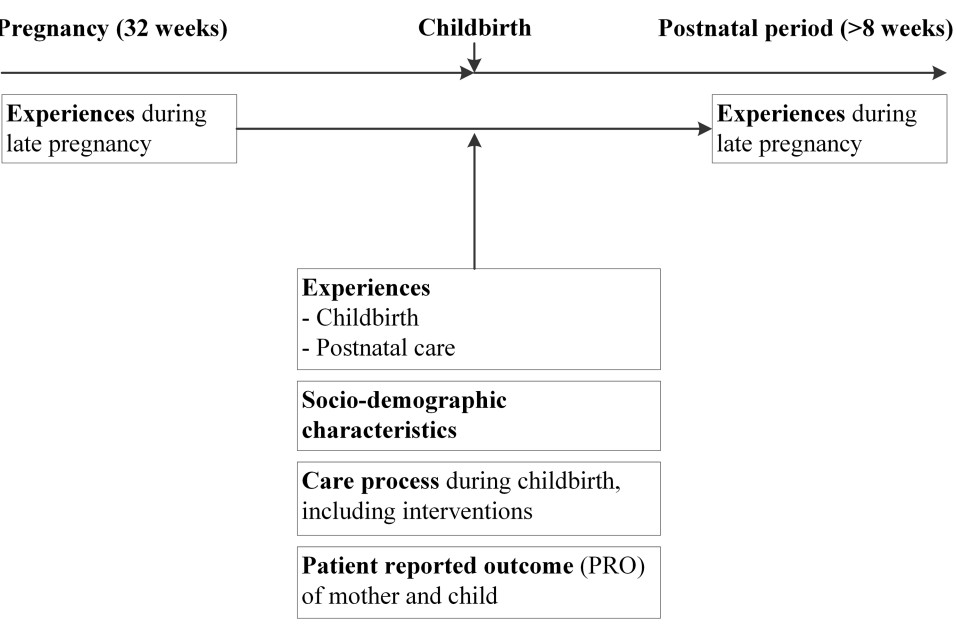

**Figure 1** Framework of analyses to determine the association of the antenatal experiences measured during pregnancy and after childbirth.

summary scores, and c. the individual domain scores, and subsequently the agreement of the gold standard and retrospective test was calculated. The agreement of the individual items between the before (gold standard) and after childbirth (retrospective test) measurement was calculated. While the domain and summary measures were calculated conventionally, for the individual item analyses, we split the 'no-agreement' category into "test better experience than retrospective test" and "test worse than retrospective test".

Second, we explored the effects of background characteristics and systematic effects of intercurrent events, as determinants of the antenatal total experience score as measured after childbirth. For the negative and median score models, we used multiple binary logistic regression analysis. For the continuous mean score model, we applied multiple linear regression analysis. Dependent variable was the antenatal total experience score as measured after childbirth; independent variables were the antenatal total experience score as measured before childbirth (gold standard score) and a set of potentially modifying factors. The following sets of determinants were included in the regression model (enter method): socio-demographic characteristics, previous experiences with care (antenatal, childbirth and postnatal care), characteristics of the care process during pregnancy and childbirth including interventions during childbirth, and perceived health outcomes of mother and child.

Considering the abundance of possible determinants and limited sample size, we included in the multivariable analyses only those that were determinants of clients' experiences during childbirth (M Scheerhagen, E Brinie, A Franx, HF Van Stel, GJ Bobsel, 2014–2015, unpublished data). A determinant was overall judged as significant if the

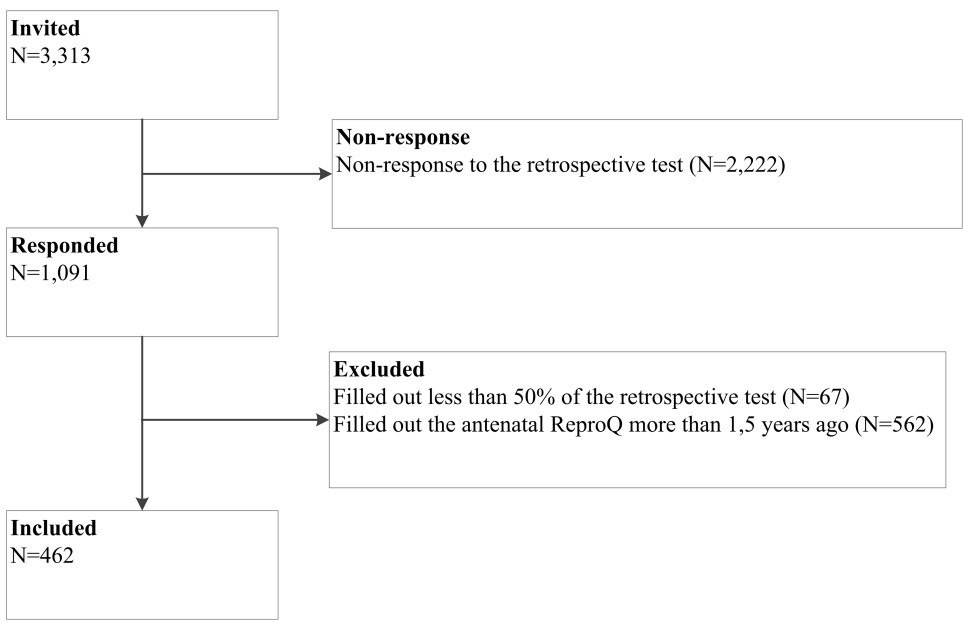

**Figure 2  Flow diagram of study.**

estimated adjusted beta- or OR-coefficient was statistically significant ($p < 0.05$, two-sided) in at least two of these analyses, a conservative approach.

For the binary logistic regression analysis, the goodness of fit was assessed using the proportion of correct predictions. For linear regression we used the adjusted $R^2$.

## RESULTS

Figure 2 shows the flow diagram. We invited 3,313 women for the retrospective test, of whom 1,091 women responded (33%). Of these, 629 women were excluded from analysis. The remaining 462 women were included.

Table 2 presents the characteristics of the included women ($n = 462$). Mean age was 32 years (SD = 4.8). Half of the women gave childbirth for the first time. 26 (6%) women were of non-Western background; and 14 (3%) women reported to have a low educational level. 241 (52%) women reported not to know the health care professional who supervised their delivery. 70 (16%) women were referred to secondary care during their pregnancy and 144 (32%) were referred during parturition. 84 (18%) women reported that they felt unhealthy and that they were hospitalized after childbirth. Additionally, 59 women (13%) perceived their babies' health as unhealthy and reported that their babies were hospitalized.

Table 3 shows the crude agreement between the antenatal experiences measured before and after childbirth for the summary and domain scores. For the total score, 35% of the women reported one or more negative experiences filling out the 'test', and 33% when filling out the retrospective test. The absolute test-retrospective test agreement (AA) of 'having a negative experience' was 67.5% (CI [63.0–71.8%]). The absolute test-retrospective test agreement (AA) of 'a score above the median' was 69.6% (CI [65.2–73.8%]). The ICC of the

**Table 2** Characteristics of women who filled out both the test and retrospective test (*n* = 462)[a].

| | | *N* | % |
|---|---|---|---|
| **Socio demographic characteristics** | | | |
| Age | ≤24 | 13 | 3 |
| | 25–29 | 130 | 28 |
| | 30–34 | 185 | 40 |
| | ≥35 | 130 | 28 |
| Parity | Primiparous | 229 | 50 |
| | Multiparous | 233 | 50 |
| Ethnic background | Western | 435 | 94 |
| | Non-Western | 26 | 6 |
| Educational level | Low | 14 | 3 |
| | Middle | 135 | 29 |
| | High | 312 | 68 |
| Marital status | Married/living together | 447 | 97 |
| | Not living together or no relationship | 14 | 3 |
| Planned pregnancy | Yes | 421 | 91 |
| | No | 41 | 9 |
| **Care process** | | | |
| Professional continuity | Yes | 220 | 48 |
| | No | 241 | 52 |
| Setting continuity | No referral | 238 | 53 |
| | Referral to secondary care during pregnancy | 70 | 16 |
| | Referral to secondary care during parturition | 144 | 32 |
| Realization of the expected place of childbirth | Yes | 263 | 58 |
| | No | 182 | 40 |
| | No prior expectations | 11 | 2 |
| **Intervention** | | | |
| Induced labor | No | 355 | 78 |
| | Yes | 103 | 23 |
| Mode of childbirth | None | 270 | 58 |
| | Episiotomy | 81 | 18 |
| | Vacuum or forceps extraction | 46 | 10 |
| | Cesarean | 65 | 14 |
| **Patient reported outcome** | | | |
| Baby | Healthy and not hospitalized | 315 | 68 |
| | Healthy, but hospitalized | 60 | 13 |
| | Unhealthy, but not hospitalized | 28 | 6 |
| | Unhealthy and hospitalized | 59 | 13 |

**Table 2** (*continued*)

|  |  | *N* | % |
|---|---|---|---|
| Mother | Healthy and not hospitalized | 245 | 53 |
|  | Healthy, but hospitalized | 27 | 6 |
|  | Unhealthy, but not hospitalized | 106 | 23 |
|  | Unhealthy and hospitalized | 84 | 18 |

**Notes.**
[a]The percentage of missing data was below 3% for all characteristics.

total experience scores (mean$_{test}$ = 3.77; mean$_{retrospective\ test}$ = 3.69) was 0.59. The negative, median and mean score models all indicated a moderate association. The associations of the personal and setting scores were comparable for the negative and median score models, but the association for the mean personal score was weaker then for the mean setting score (ICC 0.49 vs. 0.59).

All individual domains showed a good to excellent association for having a negative experience. For the median and mean scores, all domain associations were moderate, except for Confidentiality, which had an ICC of 0.27, indicating a poor association.

The item analyses showed good to excellent associations for having a negative experience (see Table 4). For the median score, the associations varied from excellent to moderate, except for 'Influence on childbirth plan' (AA = 59.7%) which was poor. For the mean score, not only this item (AA = 56.6%) but also 'Waiting time for service' (AA = 57.7%) and 'Continuity of care provision when change of professional' (across disciplines) (AA = 55.2%), had a poor association.

Table 4 also depicts the magnitude and direction of change between the before and after childbirth measurements. For the negative score, agreement was very high, indicating that scores were fairly stable between the test and retrospective test, with slightly more clients reporting negative scores at the test, the 'Birthplan' item being an exception. The median and mean scores showed more variability in scores between the test and retrospective test, with the overall trend of higher scores at the test.

Table 5 shows the results of the regression analyses. The experience score of the retrospective test were not significantly influenced by any of the socio-demographic characteristics. However, the retrospective test score was significantly associated with the women's antenatal, childbirth and postnatal experiences. Of the care process determinants, only professional continuity was relevant. Finally, the perceived maternal and infant health outcome had no significant influence on the retrospective test. Despite the different analyses and scoring models, the goodness of fit was comparable for the three measures (70–73%).

## DISCUSSION

To determine the optimal timing of the collection of data on clients' antenatal experiences, we assessed the association between the antenatal experiences measured before and after childbirth for the summary, domain and item scores. The total score showed a moderate association, irrespective of the scoring model used. For the domain scores, the associations varied with the scoring model selected, being overall excellent for the negative score, and moderate for the median and mean scores. For the domains, agreement was quite

Scheerhagen et al. (2018), *PeerJ*, DOI 10.7717/peerj.5851

**Table 3** The association between the late antenatal experiences measured during pregnancy and after childbirth, expressed as having a negative experience, below the median score and mean score (*n* = 462).

| | Negative experience score[a] | | | Median experience score[b] | | | Mean experience score | | | | |
|---|---|---|---|---|---|---|---|---|---|---|---|
| | test (%) | retrospective test (%) | Absolute agreement (AA) (%) | test (%) | retrospective test (%) | Absolute agreement (AA) (%) | Mean test | SD test | Mean retrospective test | SD retrospective test | ICC |
| **Total score** | **35.1%** | **32.5%** | **67.5%** | **60.4%** | **47.7%** | **69.6%** | **3.77** | **0.23** | **3.69** | **0.29** | **0.59** |
| **Personal score** | **22.1%** | **19.5%** | **75.8%** | **74.6%** | **59.0%** | **70.5%** | **3.81** | **0.23** | **3.72** | **0.29** | **0.49** |
| **Setting score** | **18.8%** | **19.9%** | **76.8%** | **50.2%** | **44.0%** | **69.6%** | **3.73** | **0.28** | **3.66** | **0.33** | **0.59** |
| Dignity | 2.6% | 3.5% | 96.1% | 74.0% | 58.7% | 69.9% | 3.89 | 0.24 | 3.81 | 0.31 | 0.42 |
| Autonomy | 19.9% | 15.6% | 77.5% | 75.1% | 86.0% | 74.0% | 3.64 | 0.42 | 3.61 | 0.45 | 0.42 |
| Confidentiality | 0.4% | 1.1% | 98.5% | 88.1% | 76.6% | 76.8% | 3.91 | 0.26 | 3.82 | 0.38 | 0.27 |
| Communication | 1.9% | 3.0% | 96.8% | 50.6% | 42.7% | 71.4% | 3.78 | 0.29 | 3.69 | 0.38 | 0.41 |
| Prompt Attention | 3.5% | 5.4% | 94.2% | 53.5% | 43.0% | 69.2% | 3.68 | 0.31 | 3.59 | 0.37 | 0.52 |
| Social Considerations | 1.1% | 1.5% | 97.8% | 67.3% | 64.6% | 71.9% | 3.79 | 0.35 | 3.76 | 0.41 | 0.46 |
| Basic Amenities | 2.2% | 1.9% | 96.8% | 70.1% | 59.1% | 69.6% | 3.83 | 0.32 | 3.74 | 0.39 | 0.48 |
| Choice and Continuity | 13.6% | 13.9% | 80.7% | 53.5% | 47.4% | 69.2% | 3.61 | 0.54 | 3.54 | 0.58 | 0.49 |

**Notes.**
[a]Having a negative experience (never in a domain and/or 'sometimes' in the individually chosen two most important domains).
[b]Equal or above the median.

**Table 4** Level of absolute agreement between the items measured during pregnancy and after childbirth (*n* = 462).

| Item Score | Negative experience score[a] | | | Median experience score[b] | | | Mean experience score | | |
|---|---|---|---|---|---|---|---|---|---|
| | Test = retrospective test | Test > retrospective test | Test < retrospective test | Test = retrospective test | Test > retrospective test | Test < retrospective test | Test = retrospective test | Test > retrospective test | Test < retrospective test |
| **Dignity** | | | | | | | | | |
| Respecting privacy | 99.6 | 0.4 | 0.0 | 87.4 | 10.6 | 1.9 | 87.1 | 10.7 | 2.2 |
| Treating with respect | 99.6 | 0.2 | 0.2 | 90.0 | 7.8 | 2.2 | 89.7 | 8.1 | 2.2 |
| Giving personal attention | 97.6 | 1.1 | 1.3 | 81.8 | 12.6 | 5.6 | 80.5 | 13.4 | 6.1 |
| Treating with kindness | 98.9 | 0.6 | 0.4 | 87.0 | 8.7 | 4.3 | 86.3 | 9.0 | 4.6 |
| Considering your wishes and customs | 97.6 | 1.7 | 0.6 | 77.5 | 16.0 | 6.5 | 74.8 | 18.0 | 7.2 |
| Trustworthy as health professional | 98.3 | 1.1 | 0.6 | 75.5 | 16.9 | 7.6 | 74.0 | 17.9 | 8.1 |
| **Autonomy** | | | | | | | | | |
| Refuse treatment | 96.5 | 0.6 | 2.8 | 74.2 | 15.8 | 10.0 | 69.9 | 17.4 | 12.7 |
| Involved in decision-making | 98.1 | 1.5 | 0.4 | 73.2 | 16.9 | 10.0 | 71.0 | 17.5 | 11.6 |
| Consent screening | 95.5 | 2.8 | 1.7 | 95.5 | 2.8 | 1.7 | 95.8 | 3.1 | 1.2 |
| Birthplan | 83.3 | 6.1 | 10.6 | 65.6 | 17.7 | 16.7 | 56.6 | 25.1 | 18.3 |
| **Confidentiality** | | | | | | | | | |
| Handeling your medical details and records | 100.0 | 0.0 | 0.0 | 85.5 | 9.1 | 5.4 | 85.1 | 9.6 | 5.3 |
| Secured provision of medical information to others | 98.9 | 0.9 | 0.2 | 82.0 | 14.1 | 3.9 | 80.9 | 15.2 | 3.9 |
| **Communication** | | | | | | | | | |
| Responsive to client questions | 99.6 | 0.4 | 0.0 | 83.1 | 12.6 | 4.3 | 82.4 | 13.2 | 4.3 |
| Consistency of advice across professionals | 97.8 | 1.7 | 0.4 | 68.6 | 20.6 | 10.8 | 62.7 | 24.3 | 13.0 |
| Comprehensibility of explanation | 99.6 | 0.2 | 0.2 | 82.7 | 11.5 | 5.8 | 81.9 | 12.2 | 5.9 |
| Provision of information while treated | 98.5 | 0.9 | 0.6 | 74.5 | 16.5 | 9.1 | 72.7 | 17.8 | 9.5 |
| **Prompt attention** | | | | | | | | | |
| Access for appointment/contact in urgent situations | 100.0 | 0.0 | 0.0 | 87.4 | 8.0 | 4.5 | 83.9 | 9.2 | 7.0 |
| Access for appointment/contact without urgency | 98.5 | 1.1 | 0.4 | 66.9 | 21.4 | 11.7 | 62.5 | 23.7 | 13.8 |
| Time from health care professional when requested | 99.6 | 0.4 | 0.0 | 77.3 | 15.8 | 6.9 | 75.6 | 17.4 | 7.0 |
| Waiting time for service | 95.2 | 2.8 | 1.9 | 85.9 | 10.0 | 4.1 | 57.7 | 25.7 | 16.6 |
| Setting within reach | 99.4 | 0.4 | 0.2 | 82.3 | 11.3 | 6.5 | 81.6 | 11.6 | 6.8 |
| Prompt phone response of health professional | 99.6 | 0.4 | 0.0 | 76.0 | 16.2 | 7.8 | 74.2 | 17.7 | 8.1 |
| **Social considerations** | | | | | | | | | |
| Involvement of the partner in care provision | 98.7 | 0.9 | 0.4 | 77.7 | 13.4 | 8.9 | 74.1 | 15.1 | 10.8 |
| Taking into account of family duties | 99.4 | 0.2 | 0.4 | 78.6 | 11.9 | 9.5 | 75.3 | 13.3 | 11.4 |
| Feeling supported by your family | 99.4 | 0.4 | 0.2 | 87.7 | 6.1 | 6.3 | 85.8 | 7.2 | 7.0 |

**Table 4** (*continued*)

| Item Score | Negative experience score[a] | | | Median experience score[b] | | | Mean experience score | | |
|---|---|---|---|---|---|---|---|---|---|
| | Test = retrospective test | Test > retro-spective test | Test < retro-spective test | Test = retro-spective test | Test > retro-spective test | Test < retro-spective test | Test = retro-spective test | Test > retro-spective test | Test < retro-spective test |
| **Basic amenities** | | | | | | | | | |
| Comfort of setting | 97.4 | 2.6 | 0.0 | 71.0 | 19.0 | 10.0 | 66.7 | 22.2 | 11.1 |
| Hygiene of setting | 99.1 | 0.6 | 0.2 | 84.0 | 11.3 | 4.8 | 82.6 | 12.4 | 5.0 |
| Accessibilty of setting | 99.6 | 0.2 | 0.2 | 88.7 | 7.4 | 3.9 | 88.3 | 7.6 | 4.1 |
| **Choice and continuity** | | | | | | | | | |
| Continuity of care provision when change of individual professional (same discipline) | 99.1 | 0.2 | 0.6 | 69.9 | 19.5 | 10.6 | 67.8 | 20.7 | 11.5 |
| Continuity of care provision when change of professional (across disciplines) | 97.8 | 1.5 | 0.6 | 73.2 | 18.8 | 8.0 | 55.2 | 26.4 | 18.4 |
| Allowance for selecting a preferred type of health professional | 81.6 | 8.4 | 10.0 | 73.8 | 14.5 | 11.7 | 66.8 | 17.9 | 15.3 |
| Being clear who was in charge of your care | 97.0 | 1.7 | 1.3 | 79.4 | 12.1 | 8.4 | 72.0 | 16.3 | 11.7 |

**Notes.**

[a] Having a negative experience (never in a domain and/or 'sometimes' in the individually chosen two most important domains).

[b] Equal or above the median.

Scheerhagen et al. (2018), PeerJ, DOI 10.7717/peerj.5851

**Table 5** Impact of antenatal, childbirth and postnatal experiences with care and other determinants on the total antenatal experience score measured after childbirth, according to three Scoring models (n = 462).

| Goodness of fit | Overall sign[c] | Negative experience score [a] | | | Median experience score[b] | | | Mean experience score | | |
|---|---|---|---|---|---|---|---|---|---|---|
| | | 71% | | | 73% | | | 70% | | |
| | | OR | 95% CI | p | OR | 95% CI | p | β | 95% CI | p |
| **Socio demographic characteristics** | | | | | | | | | | |
| Ethnic background | | | | | | | | | | |
| Western (ref) | | 1 | | | 1 | | | 0.00 | | |
| Non-Western | | 1.22 | 0.47 – 3.13 | 0.69 | 0.75 | 0.27 – 2.07 | 0.58 | −0.03 | −0.11 – 0.06 | 0.53 |
| Educational level | | | | | | | | | | |
| Low / middle | | 0.75 | 0.46 – 1.21 | 0.23 | 1.24 | 0.76 – 2.01 | 0.40 | −0.02 | −0.06 – 0.02 | 0.11 |
| High (ref) | | 1 | | | 1 | | | 0.00 | | |
| Planned pregnancy | | | | | | | | | | |
| Yes (ref) | | 1 | | | 1 | | | 0.00 | | |
| No | | 1.22 | 0.47 – 3.13 | 0.87 | 1.33 | 0.59 – 3.00 | 0.49 | 0.06 | −0.01 – 0.13 | 0.12 |
| **Experiences with care** | | | | | | | | | | |
| Antenatal experience | * | 3.08 | 1.95 – 4.88 | <0.01 | 3.94 | 2.51 – 6.19 | <0.01 | 0.62 | 0.52 – 0.71 | <0.01 |
| Childbirth experience | * | 2.07 | 1.32 – 3.26 | <0.01 | 1.89 | 1.16 – 3.08 | 0.01 | 0.27 | 0.17 – 0.38 | <0.01 |
| Postnatal experience | * | 1.45 | 0.89 – 2.37 | 0.14 | 2.17 | 1.35 – 3.49 | <0.01 | 0.14 | 0.05 – 0.23 | <0.01 |
| **Care process** | | | | | | | | | | |
| Professional continuity | | | | | | | | | | |
| Yes (ref) | | 1 | | | 1 | | | 0.00 | | |
| No | * | 1.60 | 0.99 – 2.60 | 0.06 | 0.50 | 0.31 – 0.82 | 0.01 | −0.05 | −0.09 – 0.00 | 0.04 |
| Setting continuity | | | | | | | | | | |
| No referral (ref) | | 1 | | | 1 | | | 0.00 | | |
| Referral during pregnancy | | 0.91 | 0.47 – 1.76 | 0.77 | 1.05 | 0.51 – 2.14 | 0.89 | 0.00 | −0.06 – 0.06 | 0.97 |
| Referral during birth | | 1.16 | 0.61 – 2.23 | 0.65 | 0.86 | 0.43 – 1.70 | 0.66 | −0.02 | −0.08 – 0.04 | 0.79 |
| Expected place of birth was realized | | | | | | | | | | |
| Yes (ref) | | 1 | | | 1 | | | 0.00 | | |
| No / no prior expectation | | 0.93 | 0.52 – 1.64 | 0.79 | 2.09 | 1.15 – 3.78 | 0.02 | 0.03 | −0.02 – 0.08 | 0.23 |
| **Intervention** | | | | | | | | | | |
| Induced labor | | | | | | | | | | |
| No (ref) | | 1 | | | 1 | | | 0.00 | | |
| Yes | | 1.50 | 0.88 – 2.55 | 0.14 | 0.77 | 0.44 – 1.35 | 0.37 | 0.02 | −0.03 – 0.07 | 0.33 |
| Intervention | | | | | | | | | | |
| No (ref) | | 1 | | | 1 | | | 0.00 | | |
| Yes | | 1.73 | 1.05 – 2.87 | 0.03 | 0.85 | 0.51 – 1.43 | 0.54 | 0.03 | −0.02 – 0.07 | 0.58 |

**Table 5** (*continued*)

| | Overall sign[c] | Negative experience score [a] | | | Median experience score[b] | | | Mean experience score | | | |
|---|---|---|---|---|---|---|---|---|---|---|---|
| **Goodness of fit** | | 71% | | | 73% | | | 70% | | | |
| | | OR | 95% CI | p | OR | 95% CI | p | β | 95% CI | | p |
| **Perceived (patient reported) outcome** | | | | | | | | | | | |
| Outcome baby | | | | | | | | | | | |
| Healthy and not hospitalized (ref) | | 1 | | | 1 | | | 0.00 | | | |
| Unhealthy and/or hospitalized | | 0.98 | 0.59 – 1.60 | 0.92 | 0.87 | 0.52 – 1.46 | 0.60 | 0.00 | −0.05 – 0.05 | | 0.99 |
| Outcome mother | | | | | | | | | | | |
| Healthy and not hospitalized (ref) | | 1 | | | 1 | | | 0.00 | | | |
| Unhealthy and/or hospitalized | | 0.89 | 0.56 – 1.43 | 0.63 | 0.81 | 0.51 – 1.30 | 0.39 | −0.03 | −0.08 – 0.01 | | 0.11 |
| **Constant** | | 0.12 | | | 0.29 | | | −0.20 | | | |

**Notes.**

[a] Having a negative experience (never in a domain and/or 'sometimes' in the individually chosen two most important domains).

[b] Equal or above the median.

[c] The determinant had a significant impact for at least two of the outcome measures.

*$p < 0.05$, two-sided.

uniform within the scoring model used. Confidentiality was the only domain with a poor association for the mean score. For the individual items, associations were particularly low for 'Influence on your childbirth plan', 'Waiting time for service', and 'Continuity of care provision when change of professional (across disciplines)'. Overall, the measurement of antenatal experiences after childbirth results in elevated variability of experiences across clients, with the overall trend that scores after birth are somewhat lower than before childbirth. Additionally, the gap between antenatal and postnatal measurement is (partly) associated with clients' experiences during childbirth and postnatal care and by professional discontinuity during childbirth, but it is unrelated to the perceived health outcome.

One key result is that the antenatal experience score measured after childbirth was only moderately associated with the antenatal experiences measured before childbirth, irrespective of the scoring model applied. In contrast, the personal, setting, domain and item scores were stronger associated for having a negative experience than for the median and mean scores. One explanation for this is that a negative experience lingers better in one's memory than an equally moderate or good experience, as shown in decision and judgment theory (*Kahneman & Tversky, 1979*; *Redelmeier & Kahneman, 1996*; *Redelmeier, Rozin & Kahneman, 1993*). An alternative explanation is of a statistical nature: changes in experiences are less easy to capture using a dichotomous measure like the negative score, producing much more agreement between the test and the retrospective test. The same argument, however, does not apply to the dichotomous median score. For the negative score, the cut-off has a fixed definition and is therefore absolute. In contrast, the cut-off for the median score equals the median of the distribution of the summary and domain scores 'as observed', and is therefore a relative position. Furthermore, the odds of having a negative experience increases with the number of items, whereas the odds of having an experience score equal or above the median is independent from the number of items.

In the ideal situation, a strong association between the antenatal experiences measured before and after childbirth is expected and desired. Furthermore, valid measurement of antenatal experiences postnatally should not be systematically affected by the care process, experiences or outcomes that occur *after* antenatal measurement. However, our results strongly suggest the opposite: women's experiences with childbirth and postnatal care had a positive and systematic impact on the antenatal experiences measured postnatally. One possibility is that women's response scales changed after birth. It is well known from research on judgment and decision (*Stiggelbout & De Vogel-Voogt, 2008*) and response shift (*Rapkin & Schwartz, 2004*; *Schwartz et al., 2007*; *Sprangers & Schwartz, 1999*), that pre-treatment judgment scales may differ systematically from post-treatment scales with, in our case, childbirth as the so-called catalyst. A change of reference frame or internal standards of comparison might result in scale recalibration (*Rapkin & Schwartz, 2004*; *Schwartz et al., 2007*; *Sprangers & Schwartz, 1999*; *Stiggelbout & De Vogel-Voogt, 2008*). The change comparison process may be related not only to a change of status quo, but also to the change of women's affect and mood after childbirth (*Stiggelbout & De Vogel-Voogt, 2008*). Another possibility is that retrospective judgment of past experiences invokes the risk of memory errors. Recall bias, i.e., 'wrong' assessment post-hoc of a former outcome (*Blome & Augustin, 2015*), may have occurred under the influence of childbirth and/or postnatal

events or experiences. Another form of memory error, so-called hindsight bias (i.e., the influence of outcome knowledge on memory reconstruction, increasing the predictability of the outcome) is less likely as (favorable) childbirth and postnatal experiences contributed positively to the gap between antenatal and postnatal measurement instead of bridging it (*Fischhoff, 2003*).

In the ideal situation, the gap between antenatal and postnatal measurement should be independent from the care process and intervention determinants. Overall, effect sizes of these variables were moderate to negligible and not significant. One exception to this is professional continuity during childbirth that was of significant impact on the antenatal experiences measured after childbirth. This is probably due, at least in part, to clients' expectations: a new professional during childbirth is never as well informed about a client's wishes and customs as her attending professional during pregnancy, and trust between the new health care professional and the client is lacking. Even though the antenatal health care professional could (and should) inform a client that a transfer during childbirth is possible, clients may not feel prepared for a change of professional.

Surprisingly, the perceived health outcome of mother and child had no impact on the antenatal experiences measured after childbirth. This is in contrast with literature, which suggests that, in retrospect, when women after childbirth recollect their antenatal experiences, these experiences could adapt in the direction of the (perceived) health outcome during childbirth; i.e., hindsight bias (*Fischhoff, 2003*; *Pohl, Bender & Lachmann, 2002*; *Ruoss, 1997*). One explanation is that hindsight bias did not occur in our case. Another explanation is that clients do not perceive a relationship between the health outcomes of childbirth and the experiences during pregnancy, as different services are provided, often by different health care professionals and often in different settings.

Another surprise is that none of the included socio-demographic determinants were significantly associated with the gap between the test and the retrospective test. This is contrary to the results of research on judgment and decision (*Stiggelbout & De Vogel-Voogt, 2008*) and response shift (*Rapkin & Schwartz, 2004*; *Schwartz et al., 2007*; *Sprangers & Schwartz, 1999*). Several explanations can be put forward. Firstly, contrary to Sprangers & Schwartz, a change of antenatal and postnatal scales (recalibration, with childbirth as the so-called catalyst) did not occur or the change was small or undetectable. Secondly, several studies suggest that the agreement between the test and retrospective test is similar between subgroups, even though the experiences are different (*Britton, 2012*; *Quintana et al., 2006*; *Raleigh et al., 2010*; M Scheerhagen, E Brinie, A Franx, HF Van Stel, GJ Bobsel, 2014–2015, unpublished data). Stated otherwise, the effect may have been cancelled within patients or even be unrelated to patient characteristics. Thirdly, the socio-demographic characteristics do not directly affect the experience scores but only exert an indirect effect, through influencing the clients' mechanisms to accommodate the change in her situation (here: childbirth) (*Rapkin & Schwartz, 2004*; *Schwartz et al., 2007*; *Sprangers & Schwartz, 1999*). Consequently, the impact of socio-demographics may already be incorporated in the impact of previous experiences. Fourthly, our sample was too small to detect any impact of socio-economic status and ethnicity on the antenatal experiences measured after childbirth. However, that argument did not apply for marital status, maternal age and parity, which

are socio-demographic characteristics that did not qualify for the multivariable analyses. Finally, we may have omitted relevant variables, e.g., personality traits or affect and mood (*Saposnik et al., 2016*; *Stiggelbout & De Vogel-Voogt, 2008*).

Our study in maternity care is a specific case of a general problem—as such it provides a warning for similar studies. Measurement problems may occur when experiences with care are evaluated but adjacent care episodes are different in terms of disease course or severity or care provided (e.g., in terms of professionals involved, locations) and separated by a critical event which could serve as 'catalyst' (e.g., intervention, hospitalization, complication). A possible change of patient's pre- and post 'catalyst' response scales and the risk of memory errors when patient's experiences are measured afterwards may result in reduced validity and/or reliability of measurements. To avoid these risks, we recommend that patient experiences with care to be measured within its own care episode.

## Strengths & limitations

One strength of this study is that, to our knowledge, this is the first study exploring the validity of clients' antenatal experiences measured after childbirth. Nevertheless, several limitations merit discussion. Firstly, women with a low educational level, non-Western women, women <24 years of age, and setting continuity (referral to secondary care) were slightly underrepresented compared to the national pregnancy population (*PRN Foundation, 2013*), despite considerable efforts to adapt the questionnaire and other measures taken to further the participation of these groups. Our results suggest, however, that these variables are all unrelated to the gap between the antenatal and postnatal measurements. Parity, induced labour, mode of delivery, and maternal and neonatal admission rates were comparable to the national average. National data on professional continuity are lacking, but data are comparable to one of our other studies ($n = 3,479$ women; M Scheerhagen, E Brinie, A Franx, HF Van Stel, GJ Bobsel, 2014–2015, unpublished data). Secondly, we did not register whether the clients' situation changed during the interval between test and retrospective test other than the events, experiences and perceptions during childbirth and postnatal care. It is possible that omitted variables could further modify the gap between test and retrospective test.

## Conclusion

Clients' experiences during pregnancy, childbirth and postnatal care are often measured for quality improvement cycles. We recommend measuring the antenatal experiences in late pregnancy instead of after childbirth, as the agreement between the antenatal experiences measured before and after childbirth is overall moderate for the summary scores.

The gap between antenatal and postnatal measurement is (partly) associated with clients' experiences during childbirth and postnatal care and by professional discontinuity during childbirth. Furthermore, measuring the antenatal experiences during pregnancy is the golden standard from a psychometric point of view. From an efficiency point of view, one could also argue to measure the antenatal experiences after childbirth and adjust the data to meet the experiences of the golden standard.

## ACKNOWLEDGEMENTS

We are grateful to Bas Henk Peter Brummelhuis for his assistance with the data collection.

### Funding
The authors received no funding for this work.

### Competing Interests
The authors declare there are no competing interests.

### Author Contributions
- Marisja Scheerhagen and Erwin Birnie conceived and designed the experiments, performed the experiments, analyzed the data, contributed reagents/materials/analysis tools, prepared figures and/or tables, authored or reviewed drafts of the paper, approved the final draft.
- Arie Franx and Henk F. van Stel contributed reagents/materials/analysis tools, prepared figures and/or tables, authored or reviewed drafts of the paper, approved the final draft.
- Gouke J. Bonsel conceived and designed the experiments, contributed reagents/materials/analysis tools, prepared figures and/or tables, authored or reviewed drafts of the paper, approved the final draft.

### Human Ethics
The following information was supplied relating to ethical approvals (i.e., approving body and any reference numbers):

The Medical Ethical Review Board, Erasmus Medical Center, Rotterdam, the Netherlands, approved the study protocol (study number MEC-2013-455).

### Data Availability
Data and the code are available in Dataverse:

https://hdl.handle.net/10411/QAYJE1.

### Supplemental Information
Supplemental information for this article can be found online at http://dx.doi.org/10.7717/peerj.5851#supplemental-information.

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
