# Peer review of "Measuring clients’ experiences with antenatal care before or after childbirth: it matters"

_PeerJ, doi:10.7717/peerj.5851_

## Round 0.1 · original submission · Minor Revisions

The manuscript has received favorable reviews

Nevertheless their comments need to be addressed before we can consider the article further for possible publication

·

Basic reporting

> Clear, unambiguous, professional English language used throughout.
The standard of English is generally good and with the exception of some minor editing will be appropriate for publication.
> Intro & background to show context.
The context and background to the key research question are well described and the need for this study justified. Please state in which countries the ReproQ is the national survey of childbirth care (Line 78).
> Literature well referenced & relevant.
Appropriate literature has been referenced.
> Structure conforms to PeerJ standards, discipline norm, or improved for clarity.
The article follows a Methods/Results/Discussion structure, including limitations and conclusions, which is appropriate for this type of report.
> Figures are relevant, high quality, well labelled & described.
The Figures are basic but generally clear.
> Raw data supplied (see PeerJ policy).
A labelled SPSS data set is provided.

Experimental design

> Original primary research within Scope of the journal.
The research topic is relevant to health service quality improvement, and can therefore be considered an aspect of health science, so although based in social science methodology it is my view that it is within the journal’s scope.
> Research question well defined, relevant & meaningful. It is stated how the research fills an identified knowledge gap.
The research addresses a clear issue in the measurement of maternal care experience, and this is important because it affects the interpretation of results from surveys of antenatal care and the future development of measurement in this area. These in turn impact on the delivery of high quality healthcare. The research question is clearly stated and defined.
> Rigorous investigation performed to a high technical & ethical standard.
The question has been carefully investigated using appropriate statistical methods, and on the whole the results are clearly reported. However, where statistical significance tests are reported (Table 4), this is done using asterisks to indicated p<0.05, whereas actual p-values should be reported. This is not only good statistical practice (see American Psychological Association style), it is also a requirement of this journal.
> Methods described with sufficient detail & information to replicate.
The methods are generally well-described and SPSS analysis syntax is provided which makes clear how analyses were conducted and will aid in replication. The ‘median’ scoring approach is a little hard to follow in the text, and could usefully be clarified by stating in the initial description that the median referred to (Line 132) is the median of all cases in the data – or if not, then to specify which reference is used. The decision to exclude women who filled out the questionnaires in paper form (Line 164) is puzzling and would benefit from some explanation.

Validity of the findings

> Negative/inconclusive results accepted.
The commentary covers positive and negative results without prejudice.
> Meaningful replication encouraged where rationale & benefit to literature is clearly stated.
There is no invitation to undertake replication of the results.
> Data is robust, statistically sound, & controlled.
The ‘funnel’ down from 3,313 women invited to take part to 462 actually participating is considerable, and presents a risk of bias. This in itself is not mentioned as a limitation of the study, although there is some comment on differences between the sample characteristics and the women originally eligible. It would be advisable to include a note about these numbers, and if possible also to provide some reassurance that other characteristics were broadly comparable.
> Speculation is welcome, but should be identified as such.
The discussion includes informed speculation, and this is clearly indicated by the wording.
> Conclusions are well stated, linked to original research question & limited to supporting results.
The conclusions are generally sound and reflect the results obtained. However, that ‘agreement between the antenatal experiences measured before and after childbirth is overall poor to moderate’ (Line 388) applies to summary scores, not necessarily to the domain scores or items (notably in relation to negative scores, when the agreement is often ‘excellent’). This should be clarified.

Additional comments

This study addresses an important aspect of the validity of survey results for antenatal care and demonstrates the impact of delaying evaluation until after delivery, as well as indicating some of the factors that may affect delayed evaluation. It has been well-conducted and the results show these findings clearly. As well as being immediately relevant to maternity care, the findings should also encourage reflection more generally on how events between a care episode and a service-user survey can impact on the ratings obtained.

Reviewer 2 ·

Basic reporting

The authors need to check that they are using the correct term in the following places:
Line 279: “Secondly, we did not administer whether the clients’ situation changed during the interval between test and retrospective test other than the events, experiences and perceptions during childbirth and postnatal care. It is possible that omitted variables could further modify the gap between test and retrospective test.”
Perhaps the word administer should be replaced with “test”.
Minor editing issues need to be addressed as shown below:
Line 66 – clarify: “events that happened since, particularly regarding the antenatal care experiences.” Since the antenatal experiences.
Line 210 – “The following sets of determinants were included (forced entry)” – meaning of “forced entry”?
Table 2 key – “(never in an domain and/or ‘sometimes’ in the individually chosen two most important domains).” ‘Never’ should also have apostrophes.
Table 2 - the “vs” –is slightly distracting and not necessary I think. Perhaps a single vertical line between the 3 types of scores could be applied instead.
Table 4- the header seems to be very long and incomplete – authors to check.

Reporting of statistics
As required by the journal, authors should report the exact p-values and not ranges.

Experimental design

To help the reader to understand the three different scoring models, given their centrality to the analysis, it may be useful to insert a small table on this could be helpful, showing the definition and descriptive statistics.

Validity of the findings

Through multiple revisions in preparation for submission of the manuscript, as inevitably happens, some discrepancies may have crept in.

In the abstract, the authors highlight two broad factors accounting for the gap between the gold standard and the retrospective assessment as follows:
“Adverse experiences during childbirth and postnatal care and lack of professional continuity during childbirth negatively influenced postnatal measurement of antenatal experiences.”

However, in searching for the phrase “adverse experiences in childbirth”, this phrase does not occur in the results, nor in the discussion. It would make it easier for the reader if there was a one-to-one alignment.
Currently the results read: “Additionally, gap between antenatal and postnatal measurement is (partly) associated with clients’ experiences during childbirth and postnatal care and by professional discontinuity during childbirth”.
In the discussion, the same observation reads: “Surprisingly, the perceived health outcome of mother and child had no impact on the antenatal experiences measured after childbirth.” - this latter description seems to contradict the statement in the abstract. Yes the effect of professional continuity is accounted for: “One exception to this is professional continuity during childbirth that was of significant impact on the antenatal experiences measured after childbirth.
In the conclusion, the related observation reads: “Measurement of antenatal
experiences postnatally is probably subject to postnatal effects.”
When reviewing Table 4 one observes that the overall rating of the experience of birth is significant in explaining the antenatal total experience score as measured after birth (dependent variable: 1.05 - 2.87 *). However, the “health outcomes” related to birth do not seem to hold explanatory power according to the regression analyses. The authors may wish to consider aligning all these statements more clearly.

Another issue is to improve the clariy of the findings in the first paragraph of the discussion. In the opening part of the discussion, state clearly what the final pattern is before indicating, as done in Line 286/7 – that “patterns were quite consistent” for the scoring model used. Then, the use and performance of different sum scores to test agreement could be compared in separate paragraphs afterwards, but only with reference to any significance of the difference in performance of the agreement statistics. In some ways I wonder whether the difference in performance of the agreement statistics distracts from the key messages. Perhaps keeping this section shorter, as long as it doesn't undermine validity, will make for a cleaner exposition.

Additional comments

I commend the authors for raising this important measurement question and resolving it with an in-depth analysis. The measurement of maternal experiences related to antenatal care is important. The manuscript supports the measurement of antenatal experience during the late stages of pregnancy as the gold standard, rather than after the birth event. Aside from being more accurate, the timing of the measurement may also assist in improving the quality cycle. Overall, the manuscript is accurate and clear, providing sufficient information on the data, methods and results. There are just minor suggestions to address the presentation of the work undertaken.

---

## Round 0.2 · accepted · Accept

The authors have addressed the commentsof the reviewers and therefore this manuscript can be accepted for publication